# From Weighting to Modeling:
# A Nonparametric Estimator for Off-Policy Evaluation

**Rong J.B. Zhu**                                           *rongzhu@fudan.edu.cn*
*Institute of Science and Technology for Brain-inspired Intelligence, Fudan University*

**Reviewed on OpenReview:** *https://openreview.net/forum?id=RW6PYOAU3w*

## Abstract

We study off-policy evaluation in the setting of contextual bandits, where we aim to evaluate a new policy using historical data that consists of contexts, actions and received rewards. This historical data typically does not faithfully represent action distribution of the new policy accurately. A common approach, inverse probability weighting (IPW), adjusts for these discrepancies in action distributions. However, this method often suffers from high variance due to the probability being in the denominator. The doubly robust (DR) estimator reduces variance through modeling reward but does not directly address variance from IPW. In this work, we address the limitation of IPW by proposing a Nonparametric Weighting (NW) approach that constructs weights using a nonparametric model. Our NW approach achieves low bias like IPW but typically exhibits significantly lower variance. To further reduce variance, we incorporate reward predictions – similar to the DR technique – resulting in the Model-assisted Nonparametric Weighting (MNW) approach. The MNW approach yields accurate value estimates by explicitly modeling and mitigating bias from reward modeling, without aiming to guarantee the standard doubly robust property. Extensive empirical comparisons show that our approaches consistently outperform existing techniques, achieving lower variance in value estimation while maintaining low bias.

## 1 Introduction

We study the off-policy value evaluation problem for decision making in environments where feedback is observed only for the actions that are taken. In this setting, the goal is to estimate the value of a target policy using data collected under an actual action-generating (behavior) policy (Langford & Zhang, 2008; Strehl et al., 2010). This problem is central to many real-world applications of reinforcement learning, especially in situations where directly deploying and evaluating a new policy is impractical due to high costs, potential risks, or ethical and legal concerns (Li et al., 2011). For example, in health care, data are typically collected under a treatment assignment policy that records outcomes only for the treatments actually administered, with no counterfactual information for treatments not received. The objective is then to evaluate the value of a new treatment policy using such observational data.

The first approach address off-policy learning in contextual bandits is inverse probability weighting (IPW) (Horvitz & Thompson, 1952), uses importance weights to correct for action imbalances in the logged data, but often incurs high variance, especially when the data logging policy has a low probability of choosing some actions. The second approach, the Direct Method (DM), which estimates the reward function from data and substitutes it as a substitute for the true reward to evaluate policy value across contexts. DM heavily relies on correct model specification and can suffer high bias when the reward model is misspecified—an issue common in practice. The third approach, doubly robust (DR) estimator (Cassel et al., 1976; Robins & Rotnitzky, 1995; Lunceford & Davidian, 2004; Kang & Schafer, 2007), combines DM and IPW and retains unbiasedness if either component is correctly specified. Although DR reduces variance through reward modeling, it does not directly address the variance introduced by the weighting mechanism itself.

In this paper, we address the high variance of the IPW approach by introducing the nonparametric weighting (NW) method, which uses a nonparametric model to link target policy-weighted rewards to behavior policy probabilities, thereby reducing the instability inherent in IPW. By employing $P$-splines to estimate this flexible function, the NW approach constructs weights that substantially reduce variance while maintaining low bias, similar to IPW. We establish convergence rates for both the bias and mean squared error of the NW estimator.

To further reduce variance, we extend our framework by incorporating reward predictions, akin to the DR technique, resulting in the Model-assisted Nonparametric Weighting (MNW) approach. We show that the MNW estimator corrects bias from the reward model and provide convergence rates for its bias and mean squared error. When applied to policy evaluation, both the NW and MNW estimators consistently outperform existing methods, achieving lower variance without sacrificing bias.

## 2 Off-policy Evaluation

Let $\mathcal{X}$ be an input space, $\mathcal{A}$ a finite action space with $K = |\mathcal{A}|$, and $\mathcal{R}$ a reward space. Consider a policy evaluation problem in contextual bandits that is specified by a distribution $\mathcal{D}$ over pairs $(x, r)$, where $x \in \mathcal{X}$ is the context and $r \in \mathcal{R}^K$ is a vector of rewards for all actions. A dataset of size $n$ is generated as follows: the environment draws a sample $(x_i, r_i)$, and the policy chooses an action $a \in \mathcal{A}$ from an unknown distribution $p$. The reward $r_{ia}$ corresponding to the context-action pair $(x_i, a)$ is then revealed. We assume

$$r_{ia} = \mu_{ia} + \epsilon_{ia},$$

where $\mu_{ia} = \mathbb{E}[r_a | x_i]$ denotes the expected reward under the context-action pair $(x_i, a)$, and $\epsilon_{ia}$ is a zero-mean noise term with finite variance $\sigma_{ia}^2$.

Two tasks are typically considered: policy evaluation and policy optimization. In policy evaluation, our goal is to evaluate the policy $\pi$. Given $x_i$, the policy $\pi$ chooses an action: $\pi_{ia} = \pi(a|x_i)$, $a \in \mathcal{A}$. We aim to estimate the value of a stationary policy $\pi$,

$$V^\pi = \mathbb{E}_{(x,r) \sim \mathcal{D}} \left[ r_{\pi(x)} \right].$$

In policy optimization, the aim is to find an optimal policy with maximum value $\pi^* = \arg\max_\pi V^\pi$. In this paper, we focus on the problem of policy evaluation. It is expected that more accurate evaluation generally leads to better policy optimization Strehl et al. (2010). Further investigation is needed into variants of this approach.

The key challenge in estimating policy value, given the data as described in the previous section, is the fact that we only have partial information about the reward, hence we cannot directly simulate our proposed policy on the data set $\mathcal{S}$. Given a data set $\mathcal{S} = \{x_i, a_i, r_{ia_i}\}_{i=1}^n$ collected as above. Define the probability associated with $x_i$ as

$$p_{ia} = \mathbb{P}(a|x_i).$$

In practice, this probability needs to be estimated, and we denote its estimator as $\hat{p}_{ia}$. Generally, there are three main approaches for overcoming this limitation.

- **DM**. The direct method forms an estimate $\hat{\mu}_{ia} = \hat{\mu}_a(x_i)$ of the expected reward on the context and action $(x_i, a)$. The policy value is then estimated by

$$\hat{V}_{\mathrm{dm}}^\pi = n^{-1} \sum_{i=1}^n \sum_{a \in \mathcal{A}} \pi_{ia} \hat{\mu}_{ia}.$$

- **IPW**. The inverse probability weighting (IPW) uses the inverse probability weighting to correct the gap between data-collection policy and the target policy:

$$\hat{V}_{\mathrm{ipw}}^\pi = n^{-1} \sum_{i=1}^n p_{ia_i}^{-1} \pi_{ia_i} r_{ia_i}.$$

- **DR**. The doubly robust (DR) approach takes advantage of both the DM and IPW approaches and constructs the following DR estimator:

$$\hat{V}_{\mathrm{dr}}^{\pi} = n^{-1} \sum\nolimits_{i=1}^{n} \left[ p_{ia_i}^{-1} \pi_{ia_i}(r_{ia_i} - \hat{\mu}_{ia_i}) + \sum\nolimits_{a \in \mathcal{A}} \pi_{ia} \hat{\mu}_{ia} \right].$$

In practice, the estimated $\hat{p}_{ia}$ is used in place of the true $p_{ia}$ in the IPW and DR approaches. The DM approach requires an accurate reward model of rewards; however, if the model is poorly specified, the DM method can incur high bias. Generally, accurately modeling rewards can be challenging, making this specification potentially restrictive. The IPW approach often suffers from large variance, particularly when the past policy differs substantially from the policy being evaluated. The DR approach improves the inference reliability by integrating the DM and IPW approaches. However, it primarily reduces variance through reward estimation, rather than specifically addressing the variance introduced by the IPW technique itself.

## 3 Nonparametric Framework of Policy Evaluation

### 3.1 Nonparametric Model Framework

In this section, we present a nonparametric framework for policy evaluation. The framework is built upon the novel representation results introduced below.

#### 3.1.1 Equivalent representations for off-policy evaluation

We now present our representation results for off-policy evaluation. Define

$$f^{\pi}(p_{ia}) = \mathbb{E}[\pi_{ia} r_{ia} | p_{ia}]. \tag{1}$$

Based on this definition, we obtain the following representation results.

**Proposition 3.1.** *Under the definition of $f^{\pi}(p_{ia})$ in Eqn. (1), the value $V^{\pi}$ of the target policy admits the following equivalent representations. (a) Design-based representation:*

$$V^{\pi} = \mathbb{E}_{x_i} \mathbb{E}_{a_i \sim p(a|x_i)} \left[ p_{ia_i}^{-1} f^{\pi}(p_{ia_i}) \right]. \tag{2}$$

*(b) Model-based representation:*

$$V^{\pi} = \mathbb{E}_{x_i} \left[ \sum\nolimits_{a \in \mathcal{A}} f^{\pi}(p_{ia}) \right]. \tag{3}$$

*Proof.* Using the unbiasedness of the IPW estimator and the definition of $f^{\pi}(p_{ia})$, we obtain

$$V^{\pi} = \mathbb{E}_{x_i} \mathbb{E}_{a_i \sim p(a|x_i)} \left[ p_{ia_i}^{-1} \pi_{ia_i} r_{ia_i} \right] = \mathbb{E}_{x_i} \mathbb{E}_{a_i \sim p(a|x_i)} \left[ \mathbb{E} \left[ p_{ia_i}^{-1} \pi_{ia_i} r_{ia_i} | p_{ia_i} \right] \right]$$
$$= \mathbb{E}_{x_i} \mathbb{E}_{a_i \sim p(a|x_i)} \left[ p_{ia_i}^{-1} f^{\pi}(p_{ia_i}) \right].$$

Thus, Eqn. (2) is proved. We also have

$$V^{\pi} = \mathbb{E}_{x_i} \left[ \sum\nolimits_{a \in \mathcal{A}} \pi_{ia} r_{ia} \right] = \mathbb{E}_{x_i} \left[ \sum\nolimits_{a \in \mathcal{A}} \mathbb{E} \left[ \pi_{ia} r_{ia} | p_{ia} \right] \right]$$
$$= \mathbb{E}_{x_i} \left[ \sum\nolimits_{a \in \mathcal{A}} f^{\pi}(p_{ia}) \right],$$

where the last step is from the definition of $f^{\pi}(p_{ia})$. Thus, Eqn. (3) is proved. $\square$

In Proposition 3.1, Eqn. (2) shows that $f^{\pi}(p_{ia})$ possesses a *design-based representation* that takes expectation with respect to the chosen action, analogous to the IPW estimator, and Eqn. (3) shows that it also admit a *model-based representation* that take expectation over all actions, analogous to the DM estimator. It is worth noting that if we instead define $f^r(p_{ia}) = \mathbb{E}[r_{ia} | p_{ia}]$, the resulting quantity $\pi_{ia} f^r(p_{ia})$ does not inherit these representation properties.

### 3.1.2 Nonparametric model framework

In the off-policy evaluation problem, the data are assumed to be collected under a mechanism in which the action assignment $a_i$ for unit $i$ relies only on $p_{ia}$. That is, conditional on $p_{ia}$, the action assignment $a_i$ is independent of $\pi_{ia} r_{ia}$. Consequently,

$$\mathbb{E}\left[\pi_{ia} r_{ia} | p_{ia}\right] = \mathbb{E}\left[\pi_{ia} r_{ia} | p_{ia}, a_i = a\right].$$

Therefore, by the definition of $f^\pi(\cdot)$ in Eqn. (1), we obtain the following model, which links $\pi_{ia} r_{ia}$ to $p_{ia}$ based on the data set $\{x_i, a_i, \pi_{ia_i} r_{ia_i}\}_{i=1}^n$, denoted by $\mathcal{S}^\pi$:

$$\pi_{ia_i} r_{ia_i} = f^\pi(p_{ia_i}) + \epsilon_{ia_i}^\pi, \ i = 1, \cdots, n, \tag{4}$$

where $\epsilon_{ia_i}^\pi$ is an error term with mean zero and variance $\sigma_{\pi, ia_i}^2$. From the proposed modeling framework (4), the function $f^\pi(\cdot)$ can be estimated from the data set $\mathcal{S}^\pi$. The function $f^\pi(\cdot)$ captures the systematic part of $\pi_{ia_i} r_{ia_i}$ that can be explained by $p_{ia_i}$, while the remainder part is absorbed by the model error term $\epsilon_{ia_i}^\pi$.

In the model framework (4), we allow for a flexible structure of $f^\pi(\cdot)$ without imposing a specific functional form, maintaining an agnostic stance toward model specification. By employing the nonparametric estimation procedure outlined in the next section, we facilitate efficient estimation of $f^\pi(\cdot)$ within this framework.

### 3.1.3 Illustration of the two cases

We now connect two weighting methods to two distinct models and demonstrate that the resulting estimators achieve optimal efficiency under their respective models.

**Case 1: IPW.** The IPW estimator can be viewed as a model-assisted estimator (Little, 2004). We construct the following working linear model to relate $r_{ia}$ to the probability $p_{ia}$. For each $(x_i, a)$, we model

$$\pi_{ia} r_{ia} = p_{ia} \beta + p_{ia} \epsilon_{ia}, \tag{5}$$

where $\{\epsilon_{ia}\}$ are assumed be independently distributed with mean 0 and variance $\sigma^2$. We estimate $\beta$ from $\mathcal{S}^\pi$. Given $\mathcal{S}^\pi$, the model follows the generalized least squares estimator $\hat{\beta}^\pi$ is given as

$$\hat{\beta}^\pi = n^{-1} \sum_{i=1}^n p_{ia_i}^{-1} \pi_{ia_i} r_{ia_i}.$$

It follows that the estimator of $\mathbb{E}[\pi_{ia} r_{ia} | p_{ia}]$ is given by $\hat{\mu}_{ia} = p_{ia} \hat{\beta}^\pi$. The corresponding estimate of $V^\pi$ is $\hat{V}^\pi = n^{-1} \sum_{i=1}^n \sum_{a \in \mathcal{A}} p_{ia} \hat{\beta}^\pi = \hat{\beta}^\pi$, where the last equality holds due to the fact $\sum_{a \in \mathcal{A}} p_{ia} = 1$ for each $i$. This demonstrates $\hat{V}^\pi = \hat{V}_{\text{ipw}}^\pi$, indicating that the IPW off-policy evaluation can be considered as a prediction under the model (5).

**Case 2: Simple Weighting.** Simple weighting could perform best under an alternative model. Consider the case which actions are taken uniformly at random regardless of the state, i.e., $p_{ia}$ is the same for every $(i, a)$. It then follows that all $f(p_{ia})$ are equal; denoted this common value by $\mu$. This leads the following model

$$\pi_{ia} r_{ia} = \mu + \epsilon_{ia}, \tag{6}$$

where $\epsilon_{ia}$ are assumed to be independently and identically distributed with mean zero and finite variance. This model means that there is no relation between $r_{ia}$ and $p_{ia}$. Given the dataset $\mathcal{S}^\pi$, the model (6) leads to the following estimator for $\mu$:

$$\hat{\mu}^\pi = n^{-1} \sum_{i=1}^n \pi_{ia_i} r_{ia_i},$$

which minimizes the variance. The corresponding value estimator is $\hat{V}_{\text{sw}}^\pi = \hat{\mu}^\pi$. This demonstrates that the simple weighting estimator is the most efficient estimator, achieving the minimum variance.

The results from the two cases above indicate that the efficiency of the IPW or SW estimators depends on which model is valid. Accordingly, modeling $\pi_{ia} r_{ia}$ through a flexible function $f^\pi(p_{ia})$ allows the estimator to capture the underlying relationship and preserve efficiency.

We illustrate how the relationship between $\pi_{ia} r_{ia}$ with the exploration policy $p_{ia}$ can conform to the function form of $f(\cdot)$.

**A Toy Example.** Suppose $r_{ia} = \mu + \delta_{ia}$, where $\delta_{ia} \sim \mathcal{N}(0, \sigma^2)$, and consider a target policy of the form $\pi_{ia} = w p_{ia} + (1-w)/K$ with $0 \le w \le 1$. Then $\pi_{ia} r_{ia} = w\mu p_{ia} + (1-w)\mu/K + \pi_{ia}\delta_{ia}$. Hence, it corresponds to a linear function $f(p_{ia}) = \alpha + \beta p_{ia}$, where $\alpha = (1-w)\mu/K$ and $\beta = w\mu$. When $w = 0$, $p_{ia}$ has no explanatory power for $\pi_{ia} r_{ia}$, in which case simple weighting is efficient; when $w = 1$, the IPW estimator fully captures their relationship. This toy example illustrates that the functional form $f(\cdot)$ captures the relationship between $\pi_{ia} r_{ia}$ and $p_{ia}$ and can be estimated using nonparametric methods.

## 3.2 Nonparametric Estimation

We provide a brief introduction to the $P$-spline approach (Eilers & Marx, 1996), which we adopt in this paper. It is worth noting that, under the general framework (4), various nonparametric methods can be employed. Here, we use the $P$-spline as a representative example due to its wide adoption in the nonparametric modeling community (Eilers & Marx, 1996). Other alternatives may also be promising, though their exploration is beyond the scope of this paper.

Consider a univariate regression model

$$y_i = f(x_i) + \zeta_i, \quad i = 1, \cdots, n,$$

where, conditionally given $x_i$, $\zeta_i$ has mean zero and variance $\sigma^2(x_i)$. We assume $f(\cdot) \in W^q[a, b]$, where $W^q[a, b]$ denotes the Sobolev space of order $q$; that is, $f(\cdot)$ has $q - 1$ absolutely continuous derivatives and satisfies $\int_a^b [f^{(q)}(x)]^2 dx < \infty$. For simplicity, we assume $x \in (0, 1)$. We adopt the $P$-spline approach (Eilers & Marx, 1996), which represents the function as

$$f(x) = \sum_{k=1}^{J(n)+d} \beta_k B_k(x),$$

where $\{B_k(x) : k = 1, \cdots, J(n) + d\}$ are $d$-th degree $B$-spline basis functions with knots $0 = \kappa_0 < \kappa_1 < \cdots < \kappa_{J(n)} = 1$. The number of knots $J(n)$ is chosen such that $J(n) = o(n)$.

The coefficient vector $\beta = (\beta_1, \cdots, \beta_{J(n)+d})^\top$ is estimated using a penalized least-squares approach. Specifically, the estimator $\hat{\beta}$ is obtained by minimizing the following objective function:

$$\sum_{i=1}^n \left[ y_i - \sum_{k=1}^{J(n)+d} \beta_k B_k(x) \right]^2 + \lambda_n \sum_{k=2}^{J(n)+d} (\Delta \beta_k)^2,$$

where $\Delta$ is the first-oder difference operator, i.e., $\Delta \beta_k = \beta_k - \beta_{k-1}$. Solving this optimization problem yields the estimator $\hat{\beta} = My$, where $M$ is the smoothing (hat) matrix resulting from the penalized least-squares procedure and $y = (y_1, \cdots, y_n)^\top$. Denoting $B(x) = (B_1(x), \cdots, B_{J(n)+d}(x))^\top$, the resulting nonparametric regression estimator of $f(x)$ is given by

$$\hat{f}(x) = B(x)^\top \hat{\beta} = B(x)^\top My. \tag{7}$$

Under regularity conditions, the penalized spline estimator is consistent; see Eubank (1999), Claeskens et al. (2009), and Xiao (2019). Specifically, for any $x \in (a, b)$, as long as $\lambda_n n^{2q-1} \to \infty$, the mean squared error satisfies $\mathbb{E}[(\hat{f}_n(x) - f(x))^2] = O(\lambda_n/n) + O(n^{1/(2q-1)} \lambda_n^{-1/(2q)})$. In particular, when $\lambda_n = O(n^{-1/(1+2q)})$, we obtain the optimal convergence rate:

$$\mathbb{E}[(\hat{f}_n(x) - f(x))^2] = O(n^{-2q/(1+2q)}). \tag{8}$$

Throughout the theoretical analysis in the paper, we adopt this optimal rate.

### 3.3 Nonparametric Weighting for Policy Evaluation

Under the model (4), we have $V^\pi = \mathbb{E}\left[\sum_{a \in \mathcal{A}} f^\pi(p_{ia})\right]$. To estimate $f^\pi(p_{ia})$ from the sample $\mathcal{S}^\pi$, we apply the nonparametric estimation described in Section 3.2. Let $y = (\pi_{1a_1} r_{1a_1}, \cdots, \pi_{na_n} r_{na_n})^\top$. Using the $P$-spline approach in (7), we obtain the estimator $\hat{f}^\pi(p_{ia}) = y^\top w_{\mathrm{nw}}(p_{ia})$, which is linear in $y$. Here, the weight vector $w_{\mathrm{nw}}(p_{ia}) = B(p_{ia})^\top M$ is derived from the penalized least-squares estimator presented in (7). We then estimate $V^\pi$ by

$$\hat{V}_{\mathrm{nw}}^\pi = n^{-1} \sum_{i=1}^n \sum_{a \in \mathcal{A}} \hat{f}^\pi(p_{ia}). \tag{9}$$

This reveals that $\hat{V}_{\mathrm{nw}}^\pi$ is a weighted average of the elements in $\{\pi_{ia_i} r_{ia_i}\}_{i=1}^n$, with the overall weight vector $n^{-1} \sum_{a \in \mathcal{A}} w_{\mathrm{nw}}(p_{ia})$. We refer to this approach as *Nonparametric Weighting* (NW).

The procedure is summarized in Algorithm 1 below.

---

**Algorithm 1** Policy Evaluation using the Nonparametric Weighting approach

---

**Input:** Policy $\pi$ to be evaluated and the values of $p_{ia_i}$ (or their estimates thereof obtained from the collected data).
**Step 1:** Apply the $P$-spline approach in (7) to regress $\{\pi_{ia_i} r_{ia_i}\}_{i=1}^n$ on $\{p_{ia_i}\}_{i=1}^n$, and obtain the fitted function $\hat{f}^\pi(\cdot)$;
**Step 2:** Compute the estimate $\hat{V}_{\mathrm{nw}}^\pi$ according to (9).

---

### 3.4 Robustness to Behavior Policy Estimation

In practice, the true values of $p_{ia}$ are typically unknown and must be estimated from the data. To address this, the function $f(\cdot)$ is subsequently fitted using the estimates $\hat{p}_{ia}$. We now examine the implications of using $\hat{p}_{ia}$, accounting for both the estimation error and bias arising from potential misspecification in the estimation of $p_{ia}$. Denote $p_{ia}^* = \mathbb{E}[\hat{p}_{ia}]$ and write $\hat{p}_{ia} = p_{ia}^* + o_p(1)$. The difference $p_{ia}^* - p_{ia}$ captures the bias due to potential model misspecification for estimating $p_{ia}$, while the term $o_p(1)$ represents estimation error.

First, we investigate the impact of estimation error. Consider no-bias case, that is, $p_{ia}^* = p_{ia}$. Assuming that $f(\cdot)$ is Lipschitz continuous, we have

$$f(\hat{p}_{ia}) = f(p_{ia}) + o_p(1).$$

Hence, evaluating the function at the estimated probabilities is asymptotically equivalent to evaluating it at the true probabilities, up to a negligible error term. This indicates that our approach remains valid and is robust to estimation error in the logging policy.

Second, we investigate the impact of bias arising from model specification. Suppose that $p_{ia}^*$ is a transformation of $p_{ia}$ through a mapping $h(\cdot)$, i.e., $p_{ia}^* = h(p_{ia})$. Although $h(p_{ia})$ is a biased proxy for $p_{ia}$, using a flexible regression model – such as a nonparametric method – allows us to approximate $\mathbb{E}[\pi_{ia} r_{ia} \mid h(p_{ia})]$. If $h(p_{ia})$ is sufficient for $p_{ia}$ with respect to $\pi_{ia} r_{ia}$, namely, $\mathbb{E}[\pi_{ia} r_{ia} \mid h(p_{ia})] = \mathbb{E}[\pi_{ia} r_{ia} \mid p_{ia}]$, then modeling based on $p_{ia}$ or $h(p_{ia})$ yields equivalent results. A simple example of this equivalence is that $h(p)$ is one-to-one function. More generally, when the biased proxy $h(p_{ia})$ is not exactly sufficient but is highly correlated with $p_{ia}$, the resulting fit can still provide a reasonable approximation to the true conditional mean. In this cases, the flexibility of the regression function mitigates the impact of the bias, rendering its effect on the final estimator limited.

### 3.5 Error Analysis

We now analyze the error bounds of the NW estimator $\hat{V}_{\mathrm{nw}}^\pi$ relative to $V^\pi$ to address the question: How well does this estimator perform? Let $\Delta_f^\pi(p) = \hat{f}^\pi(p) - f^\pi(p)$, and define $\bar{V}^\pi = n^{-1} \sum_{i=1}^n \sum_{a \in \mathcal{A}} f^\pi(p_{ia})$ as the

sample average of $f^{\pi}(p_{ia})$. Then, the error of the NW estimator can be decomposed as

$$
\begin{aligned}
\hat{V}_{\mathrm{nw}}^{\pi} - V^{\pi} =& \hat{V}_{\mathrm{nw}}^{\pi} - \bar{V}^{\pi} + \bar{V}^{\pi} - V^{\pi} \\
=& n^{-1} \sum_{i=1}^{n} \sum_{a \in \mathcal{A}} \Delta_{f}^{\pi}(p_{ia}) + n^{-1} \sum_{i=1}^{n} \sum_{a \in \mathcal{A}} f^{\pi}(p_{ia}) - V^{\pi}.
\end{aligned} \tag{10}
$$

Noting $\mathbb{E}[(\Delta_{f}^{\pi}(p_{ia}))^2] = O(n^{-2q/(1+2q)})$, as established in Eqn. (8), and $\mathbb{E}[(\bar{V}^{\pi} - V^{\pi})^2] = O(n^{-1})$, we see from Eqn. (10) that the uncertainty in $\hat{V}_{\mathrm{nw}}^{\pi}$ relative to $V^{\pi}$ mainly stems from $\Delta_{f}^{\pi}(p_{ia})$, which captures the estimation error in the nonparametric procedure. The following proposition establishes the convergence rates for the bias and mean squared error of $\hat{V}_{\mathrm{nw}}^{\pi}$.

**Proposition 3.2.** *(a) The bias of the NW estimator is given by*

$$
\mathbb{E}(\hat{V}_{nw}^{\pi}) - V^{\pi} = O(Kn^{-q/(1+2q)}).
$$

*(b) The MSE of the NW estimator is given by*

$$
\mathbb{E}[(\hat{V}_{nw}^{\pi} - V^{\pi})^2] = O(K^2 n^{-2q/(1+2q)}).
$$

Proposition 3.2 shows that the convergence rates depend on both $K$ and $n$, under the condition that $Kn^{-q/(1+2q)} \to 0$ as $n \to \infty$. As a result, convergence is guaranteed even in the presence of a large action space, provided that $K = o(n^{q/(1+2q)})$.

### 3.6 An Illustrative Example

We present Example 1 to demonstrate how our proposed NW method can improve efficiency. We simulate a $K$-armed bandit setting with 300 contexts. For each context $i$ and arm $k = 1, \cdots, K$, $y_{ik}$ is independently generated from $\mathcal{N}(0,1)$, and we consider the squared rewards $y_{ik}^2$. We construct three distinct reward-generating scenarios:

- Sorting $\{y_{i1}^2, \cdots, y_{iK}^2\}$ in increasing order;
- Sorting $\{y_{i1}^2, \cdots, y_{iK}^2\}$ in decreasing order;
- Leaving $\{y_{i1}^2, \cdots, y_{iK}^2\}$ unsorted.

For each context $i$, we draw $K$ values from a uniform distribution over $(0,1)$ and sort them in increasing order. An arm is then sampled with probability proportional to these sorted values.

Sorting the rewards in increasing order induces a strong positive correlation between the rewards and the sampling probabilities, while sorting them in decreasing order induces a strong negative correlation. In such cases, an approximate working model can exploit this explanatory power through nonparametric modeling of the probabilities. In contrast, when the rewards remain unsorted, there is no correlation between the rewards and the sampling probabilities. In this case, the probabilities offer no additional explanatory power, and the SW estimator yields the most efficient evaluation.

We set $K = 20$ and obtain a sample of size $n = 300$. Our goal is to evaluate the uniform target policy $\pi_k = 1/K$, leading to the objective of estimating $V = (nK)^{-1} \sum_{i=1}^{n} \sum_{k=1}^{K} y_{ik}^2$. We estimate $V$ using the SW and IPW estimators, as well as the proposed NW estimator. With $B = 2000$ iterations, we calculate the bias, standard deviation (s.d.), and root mean square error (RMSE) for each estimator. The results are presented in Table 1. From the table, the variance of the NW estimator is significantly smaller than that of the IPW estimator across all three cases, resulting in much higher efficiency in terms of RMSE, while the NW estimator exhibits slightly larger absolute bias compared to the IPW estimator. Compared to the SW estimator, which has the lowest variance in the unsorted and decreasing cases but the largest absolute bias, the NW estimator performs much better in the decreasing and increasing cases while being slightly worse in the unsorted case, as expected.

Table 1: Performance of Example 1

| method | decreasing | | | increasing | | | unsorted | | |
|---|---|---|---|---|---|---|---|---|---|
| | Bias | s.d. | RMSE | Bias | s.d. | RMSE | Bias | s.d. | RMSE |
| SW | -0.586 | **0.040** | 0.587 | 0.577 | 0.090 | 0.584 | 0.018 | **0.081** | **0.083** |
| IPW | **0.017** | 0.491 | 0.492 | **-0.002** | 0.045 | 0.045 | **0.004** | 0.258 | 0.258 |
| NW | -0.070 | 0.148 | **0.164** | 0.025 | **0.031** | **0.040** | 0.007 | 0.110 | 0.110 |

## 4 Model-assisted Nonparametric Weighting

In this section, we extend the NW approach by integrating the DM estimator and propose a model-assisted nonparametric weighting (MNW) estimator. Given an estimate $\hat{\mu}_{ia}$ of the expected reward for the context-action pair $(x_i, a)$, we compute the residual as $\pi_{ia}(r_{ia} - \hat{\mu}_{ia})$. We model the relationship between $\pi_{ia}(r_{ia} - \hat{\mu}_{ia})$ and the probability $p_{ia}$ using the following nonparametric model for each $(x_i, a)$:

$$\pi_{ia}(r_{ia} - \hat{\mu}_{ia}) = g^\pi(p_{ia}) + \xi_{ia}^\pi, \tag{11}$$

where $\{\xi_{ia}^\pi\}$ are independently distributed with mean 0 and variance $\nu_{\pi,ai}^2$.

Similar to Section 3.3, we estimate the function $g^\pi(\cdot)$ from the sample $\mathcal{S}^\pi$ and obtain the nonparametric estimate $\hat{g}^\pi(\cdot)$ under the model (11). Given $\hat{g}^\pi(\cdot)$, we construct the MNW estimator as

$$\hat{V}_{\mathrm{mnw}}^\pi = n^{-1} \sum_{i=1}^n \sum_{a \in \mathcal{A}} (\hat{g}_{ia}^\pi + \pi_{ia}\hat{\mu}_{ia}). \tag{12}$$

Like the DR estimator, the MNW estimator incorporates $\hat{\mu}_{ia}$ as a baseline when a reward model is available, reducing variance by nonparametrically modeling the residuals $r_{ia} - \hat{\mu}_{ia}$. At the same time, this incorporation does not introduce bias. In other words, the MNW estimator remains robust to misspecification of $\hat{\mu}_{ia}$: when $\hat{\mu}_{ia}$ is biased, $\hat{V}_{\mathrm{mnw}}^\pi$ compensates through the nonparametric adjustment provided by $\hat{g}^\pi(\cdot)$, which captures and corrects the residual bias. As a result, the MNW estimator effectively adjusts for misspecification in the reward model $\mu_a(\cdot)$, achieving higher efficiency when $\hat{\mu}_a(\cdot)$ is accurate and maintaining robustness when it is not. The procedure is summarized in Algorithm 2 below.

---

**Algorithm 2** Policy Evaluation using the Model-assisted Nonparametric Weighting approach

---

**Input:** Policy $\pi$ to be evaluated and the values of $p_{ia_i}$ (or their estimates obtained from the collected data).
**Step 1:** Estimate $\mu_{ia}$ and obtain the fitted values $\hat{\mu}_{ia}$;
**Step 2:** Apply the $P$-spline approach to regress $\{\pi_{ia_i}(r_{ia_i} - \hat{\mu}_{ia_i})\}_{i=1}^n$ on $\{p_{ia_i}\}_{i=1}^n$, yielding the fitted function $\hat{g}_{ia}^\pi(\cdot)$;
**Step 3:** Compute the estimate $\hat{V}_{\mathrm{mnw}}^\pi$ according to (12).

---

### 4.1 Error Analysis

We now analyze the bias and MSE of the MNW estimator. Denoting $\mu_{ia}^* = \mathbb{E}(\hat{\mu}_{ia})$, the bias introduced by the reward model is given by $\mu_{ia} - \mu_{ia}^*$. Under the nonparametric model (11), we have

$$\mathbb{E}[\pi_{ia}(r_{ia} - \hat{\mu}_{ia})] = \pi_{ia}(\mu_{ia} - \mu_{ia}^*) = g^\pi(p_{ia}). \tag{13}$$

Define $\bar{V}_{\mathrm{mnw}}^\pi = n^{-1} \sum_{i=1}^n \sum_{a \in \mathcal{A}} \pi_{ia}\mu_{ia}$. Using (13), we obtain

$$\bar{V}_{\mathrm{mnw}}^\pi = n^{-1} \sum_{i=1}^n \sum_{a \in \mathcal{A}} (\pi_{ia}\mu_{ia}^* + g^\pi(p_{ia})). \tag{14}$$

Let $\Delta_g^\pi(p) = \hat{g}^\pi(p) - g^\pi(p)$. From (14), we have $\hat{V}_{\mathrm{mnw}}^\pi - \bar{V}_{\mathrm{mnw}}^\pi = \Delta_g^\pi(p_{ia}) + (\hat{\mu}_{ia} - \mu_{ia}^*)$. This leads to the following decomposition:

$$\hat{V}_{\mathrm{mnw}}^\pi - V^\pi = \hat{V}_{\mathrm{mnw}}^\pi - \bar{V}_{\mathrm{mnw}}^\pi + \bar{V}_{\mathrm{mnw}}^\pi - V^\pi$$
$$= n^{-1} \sum_{i=1}^n \sum_{a \in \mathcal{A}} \left[ \Delta_g^\pi(p_{ia}) + (\hat{\mu}_{ia} - \mu_{ia}^*) \right] + n^{-1} \sum_{i=1}^n \left[ V_{\mathrm{mnw}}^\pi(i) - V^\pi \right]. \qquad (15)$$

where $V_{\mathrm{mnw}}^\pi(i) = \sum_{a \in \mathcal{A}} \left( \pi_{ia} \mu_{ia}^* + g^\pi(p_{ia}) \right)$.

Based on this decomposition, we derive the convergence rates for the bias and mean squared error of $\hat{V}_{\mathrm{mnw}}^\pi$, as stated in the following proposition.

**Proposition 4.1.** *Assume* $\mathbb{E}[(\hat{\mu}_{ia} - \mu_{ia}^*)^2] = O(n^{-1})$. *Then, we have the following results: (a) The bias of the MNW estimator is given by*

$$\mathbb{E}(\hat{V}_{mnw}^\pi) - V^\pi = O(K n^{-q/(2q+1)}).$$

*(b) The MSE of the MNW estimator is given by*

$$\mathbb{E}[(\hat{V}_{mnw}^\pi - V^\pi)^2] = O(K^2 n^{-2q/(2q+1)}).$$

Proposition 4.1 demonstrates that the MNW estimator remains consistent despite discrepancies in reward modeling (i.e., $\mu_{ia} - \mu_{ia}^*$), as these discrepancies are corrected by $g^\pi(\cdot)$.

**Comparison with the NW estimator.** Eqn. (15) shows that $\hat{V}_{\mathrm{mnw}}^\pi - \bar{V}_{\mathrm{mnw}}^\pi$ can be decomposed into two components: $\Delta_g^\pi(p_{ia})$, which reflects the error in estimating $g^\pi(\cdot)$, and $\hat{\mu}_{ia} - \mu_{ia}^*$, the deviation of the estimated reward from its expectation. When the variability of $\hat{\mu}_{ia} - \mu_{ia}^*$ is smaller than that of $\Delta_g^\pi(p_{ia})$, the dominant source of error arises from $\Delta_g^\pi(p_{ia})$. Comparing $\Delta_g^\pi(p_{ia})$ in the MNW estimator with $\Delta_f^\pi(p_{ia})$ in the NW estimator, it becomes evident that the MNW estimator can achieve higher efficiency when $\pi_{ia} \hat{\mu}_{ia}$ effectively captures context-dependent reward information.

## 4.2 An Illustrative Example

We present Example 2 to illustrate how the MNW approach can improve efficiency over the NW approach. The setting is similar to that in Section 3.6, but the reward-generating process differs. Specifically, we generate $z_{ik} = x_{ik}^2 + y_{ik}^2$, where $x_{ik} \sim \mathcal{N}(0, 2)$ and $y_{ik} \sim \mathcal{N}(0, 1)$ are independently generated. Here, the term $x_{ik}^2$ represents the baseline reward. We adopt the baseline reward model $\mu(x) = \beta x^2$, with $\beta = 1$ corresponding correct specification and $\beta = 0.5$ corresponding to misspecification. As in Example 1, we construct three reward-generating scenarios by sorting $\{z_{ik}\}_{k=1}^K$ for each $i$ according to the values of $\{y_{ik}^2\}_{k=1}^K$. We simulate a 20-armed bandit setting with 300 contexts, where rewards are independently generated across contexts and arms. The corresponding results are presented in Table 2.

Table 2: Performance of Example 2

| method | decreasing | | | increasing | | | unsorted | | |
|---|---|---|---|---|---|---|---|---|---|
| | Bias | s.d. | RMSE | Bias | s.d. | RMSE | Bias | s.d. | RMSE |
| SW | -0.604 | **0.157** | 0.624 | 0.553 | 0.177 | 0.581 | -0.036 | **0.174** | **0.178** |
| IPW | **0.005** | 0.693 | 0.693 | -0.002 | 0.345 | 0.345 | 0.007 | 0.444 | 0.444 |
| NW | -0.148 | 0.259 | 0.298 | -0.003 | 0.243 | 0.243 | **0.000** | 0.265 | 0.265 |
| MNW($\beta = 0.5$) | -0.118 | 0.220 | 0.250 | 0.003 | 0.184 | 0.184 | -0.013 | 0.213 | 0.213 |
| MNW($\beta = 1$) | **-0.080** | 0.221 | **0.235** | **0.000** | **0.157** | **0.157** | -0.025 | **0.194** | **0.195** |

From the table, the MNW estimator exhibits substantially lower variance than the NW estimator across all three cases, resulting in higher efficiency in terms of RMSE, while maintaining negligible bias comparable to that of the NW estimator. The MNW estimator works better when $\beta = 1$ than when $\beta = 0.5$. Note that $\beta = 0.5$ corresponds to a misspecified baseline model, in which only part of the term $x_{ik}^2$ is explained by the model. The results show that the MNW estimator still performs better, as expected, since the nonparametric

weighting approach efficiently mitigates bias arising from model misspecification. This example illustrates that the MNW estimator can improve performance when the baseline model captures some of the outcome variation, in a manner analogous to doubly robust estimation.

## 5  Experiments

This section provides empirical evidence supporting the effectiveness of NW-type estimators over IPW-type estimators. Although more recent methods for off-policy evaluation have been developed (see Section 6 below), we restrict our comparison to the IPW and DR estimators. These estimators are conceptually representative and sufficient to highlight the essential differences between weighting-based and nonparametric approaches. Including a larger set of methods would not necessarily provide additional insights, as our primary aim is to examine the relative performance and robustness of the NW-type estimators in comparison with the standard IPW framework.

Following Dudík et al. (2011), we consider a multi-class classification problem with bandit feedback, using public benchmark datasets. We use the same datasets as in Dudík et al. (2011). See Table 6 in the appendix for a summary of using the datasets used. All experiments were conducted on a MacBook Air equipped with M3.

In a classification task, we assume that the data are drawn independently and identically distributed from a fixed distribution: $(x, c) \sim P$, where $x \in \mathcal{X}$ is the feature vector and $c \in \{1, \cdots, K\}$ is the class label. The goal is to find a classifier $\pi : \mathcal{X} \to \{1, \cdots, K\}$ that minimizes the classification error $e^\pi = \mathbb{E}_P[I(\pi(x) \neq c)]$. We turn the data point $(x, c)$ into a cost-sensitive classification example $(x, l_1, \cdots, l_K)$, where $l_a$ denotes the loss for predicting $a$. We consider a noisy loss where $l_a$ is drawn from a Gaussian distribution with mean $I(a \neq c)$ and $\sigma = 0.2$. Under this formulation, the classifier $\pi$ can be viewed as an action-selection policy, and its classification error is exactly the policy's expected loss. The target policy $\pi$ is defined as the deterministic decision of a logistic regression classifier learned on the multi-class data.

The logging policy $b$ is generated by sampling probability proportional to values drawn from a uniform distribution; specifically, we draw $p_a \sim \text{unif}(0, 1)$ and assign action $a$ according to $b_a \propto p_a$. To predict the value for action $a$ for unit $i$ in the testing dataset, we first fit an $l_2$-regularized least-squares estimates using the training data, and then generate predictions for each data point in the test set.

We randomly split data into training and test sets of (roughly) the same size. On the training set, we train a multinomial classifier to define the policy $\pi$, and compute the classification error on the test data, treating it as the ground truth for comparing various estimates. We perform Monte Carlo simulation with 500 iterations to generate a partially labeled set from the test data using the logging policy $b$. We compute policy evaluation estimates using various methods and calculate the resulting bias and root mean squared error (RMSE). We repeat this entire process 20 times, and report the average bias and RMSE values across these 20 runs in Table 3. The data sizes used for policy evaluation vary across datasets (see Table 6 in the appendix for a summary). We further examine the performance under varying sample sizes, and report results for a representative dataset, Page, in Figure 1 in the appendix. The results demonstrate that our approaches are robust to the sample size used for policy evaluation.

From the table, the RMSE of the NW estimator is consistently lower than that of the IPW estimator across all datasets, while its bias is remains negligible and comparable to that of the IPW estimator. The DM estimator performs the worst, reflecting the poor fit of the linear model used to predict rewards. In comparison, the MNW estimator achieves substantially lower RMSE than the DR estimator, while maintaining a similar level of bias.

**How much does estimating the logging policy affect the results?**  To assess the impact of estimating the logging policy $b$, we conduct another experiment using a perturbed logging policy defined as $\tilde{b} \propto p_a * \delta$, where $\delta \sim \mathcal{N}(1, 0.09)$. Here, $\delta$ introduces Gaussian noise into the probability estimates, simulating errors in estimating the logging policy that generated the data. We calculate the bias and RMSE values of each estimator using $\tilde{b}$ and report the results in Table 4. As shown in the table, the IPW and DR estimators exhibit significantly larger RMSE, highlighting their sensitivity to errors in probability estimation. Notably, the IPW estimator shows a marked increase in bias, suggesting that it may become biased under noisy probability

Table 3: Bias and RMSE of various estimators for classification error with the true logging policy.

| Data | Bias | | | | | RMSE | | | | |
|------|------|-----|-----|-----|-----|------|-----|-----|-----|-----|
| | DM | IPW | DR | NW | MNW | DM | IPW | DR | NW | MNW |
| letter | 0.507 | 0.001 | 0.009 | **0.000** | 0.009 | 0.507 | 0.070 | 0.075 | **0.036** | 0.045 |
| glass | 0.218 | -0.003 | 0.034 | **0.000** | 0.036 | 0.218 | 0.233 | 0.238 | 0.238 | **0.193** |
| ecoli | 0.215 | 0.002 | 0.037 | **0.000** | 0.034 | 0.215 | 0.270 | 0.229 | 0.243 | **0.208** |
| opt | 0.292 | 0.001 | -0.001 | **0.000** | -0.001 | 0.292 | 0.047 | 0.060 | **0.027** | 0.037 |
| page | 0.086 | **0.000** | 0.012 | -0.001 | 0.012 | 0.086 | 0.021 | 0.028 | **0.014** | 0.020 |
| pen | 0.400 | 0.000 | 0.004 | **0.000** | 0.004 | 0.400 | 0.033 | 0.051 | **0.022** | 0.032 |
| sat | 0.192 | 0.000 | 0.020 | **0.000** | 0.021 | 0.192 | 0.041 | 0.043 | **0.024** | 0.031 |
| vehicle | 0.239 | 0.001 | 0.017 | **0.000** | 0.016 | 0.239 | 0.079 | 0.078 | 0.058 | **0.057** |
| yeast | 0.215 | **0.000** | 0.070 | -0.002 | 0.070 | 0.215 | 0.139 | 0.140 | **0.098** | 0.106 |

Table 4: Bias and RMSE of various estimators for classification error under perturbed logging policy.

| Data | Bias | | | | | RMSE | | | | |
|------|------|-----|-----|-----|-----|------|-----|-----|-----|-----|
| | DM | IPW | DR | NW | MNW | DM | IPW | DR | NW | MNW |
| letter | 0.505 | 0.066 | -0.085 | **0.000** | 0.009 | 0.505 | 0.319 | 0.337 | **0.034** | 0.041 |
| glass | 0.215 | 0.044 | 0.015 | **-0.002** | 0.039 | 0.215 | 0.302 | 0.282 | 0.226 | **0.188** |
| ecoli | 0.206 | 0.060 | 0.012 | **0.002** | 0.039 | 0.206 | 0.334 | 0.272 | 0.221 | **0.190** |
| opt | 0.291 | 0.006 | -0.052 | **0.000** | -0.001 | 0.291 | 0.057 | 0.115 | **0.025** | 0.035 |
| page | 0.088 | 0.004 | -0.001 | **0.000** | 0.012 | 0.088 | 0.043 | 0.048 | **0.013** | 0.019 |
| pen | 0.398 | 0.020 | -0.095 | **0.000** | 0.004 | 0.398 | 0.081 | 0.216 | **0.020** | 0.030 |
| sat | 0.192 | 0.028 | -0.007 | **-0.002** | 0.021 | 0.192 | 0.074 | 0.085 | **0.023** | 0.030 |
| vehicle | 0.236 | 0.021 | -0.027 | **-0.001** | 0.015 | 0.236 | 0.126 | 0.164 | 0.054 | **0.053** |
| yeast | 0.218 | 0.044 | -0.043 | **-0.003** | 0.069 | 0.218 | 0.230 | 0.470 | **0.089** | 0.101 |

estimates. In contrast, the NW and MNW estimators yield RMSE values similar to those obtained when the logging policy is known, demonstrating that our proposed methods are robust to inaccuracies in the estimated probabilities. We further conduct an experiment in which the logging policy $b$ is estimated; the corresponding performance results are reported in Table 5 of the appendix, The results show that our approaches perform consistently across settings.

## 6 Related Work

Off-policy evaluation for bandits has been extensively studied. Below, we provide a concise review of the most relevant work to the best of our knowledge.

To address the high variance of IPW-based methods, several simple techniques have been proposed. One common approach is weight clipping, which truncates large inverse probability weights (Ionides, 2008; Swaminathan & Joachims, 2015a; Su et al., 2019). Another is normalization, which rescales the weights to stabilize estimates (Swaminathan & Joachims, 2015b). In contrast, our paper takes a fundamentally different approach by modeling the probabilities directly, thereby avoiding the issue of large weights in a data-driven way.

Doubly robust estimation (Robins et al., 1994; Lunceford & Davidian, 2004; Kang & Schafer, 2007) is widely used for parameter estimation in statistical inference. Dudík et al. (2011) first applied this framework to policy evaluation and optimization, with later extensions to the reinforcement learning setting by Jiang & Li (2016) and Thomas & Brunskill (2016). More recent developments include the work of Wang et al. (2017), Farajtabar et al. (2018), and Su et al. (2020). While the goal of our MNW approach is not to guarantee the standard doubly robust property, it explicitly models and mitigates the bias introduced by reward modeling.

Doubly robust estimation has also been explored in observational settings for estimating average treatment effects using asymptotically optimal estimators (Hirano et al., 2003; Imbens et al., 2007).

Several data-driven approaches have been proposed for off-policy evaluation. Saito et al. (2021) selected one of multiple logging policies as a pseudo-target policy, directly estimated its value from the dataset, and used it to identify the off-policy estimator with the best performance. Udagawa et al. (2023) introduced two surrogate policies constructed from the logged data. Cief et al. (2024) proposed a cross-validated off-policy evaluation framework. In contrast to these methods, our approach leverages flexible modeling, offering both ease of implementation and reliable performance.

Finally, it is worth noting that nonparametric modeling of sampling probabilities is not new. In survey sampling, such techniques have been employed to correct for sample selection bias and enable finite population inference; see the review article (Little, 2004). Motivated by this line of work, we propose a nonparametric wighting approach for policy evaluation, and further enhance it by incorporating the regression-based adjustment to improve accuracy.

## 7 Conclusion

In this paper, we have proposed nonparametric modeling for off-policy evaluation, leading to nonparametric weighting estimators for off-policy evaluation. The weights are constructed using a nonparametric framework rather than relying on explicit selection bias adjustment. Instead, the approach is grounded in the general model specification (4) and leverages data-driven relationships. The method exhibits strong robustness due to its flexible modeling structure. When the rewards are only weakly correlated with the selection probabilities, there is little relationship for the nonparametric model to capture; in such cases, the NW estimator essentially reduces to simple averaging. As the relationship becomes stronger, the nonparametric model is able to capture more of the underlying structure, thereby improving its explanatory and predictive power. Furthermore, we incorporate the DM estimator to further reduce the variance in the rewards and develop a model-assisted nonparametric weighting estimator. Extensive experiments on a multi-class classification with bandit feedback, using public benchmark datasets, demonstrate the efficacy of these estimators and emphasize the role of nonparametric weighting in achieving superior performance. As a result, we anticipate that the NW approach will become standard alternatives to the IPW approach.

This study has some limitations. First, we adopt the $P$-spline approach to construct the NW-type estimators. Although the $P$-spline method demonstrates robust performance in our experiments, implementing the framework with alternative nonparametric methods, particularly neural networks models, could further enhance its flexibility and predictive accuracy. This direction warrants further investigation. Second, in many reinforcement learning problems, the rewards are specified as binary variables. However, the general model framework (4) does not explicitly account for the discrete nature of such reward distributions. Incorporating the discrete characteristics of the outcome may further improve the performance and applicability of the proposed approach.

Several promising directions for further research arise from this work. Given the widespread use of IPW in policy evaluation across various domains, we anticipate that our NW approach will have competitive applications in these areas. For instance, extending it to combinatorial contextual bandits (Swaminathan et al., 2017) could offer valuable insights, particularly in decision-making scenarios with complex action spaces. Furthermore, integrating this methodology within the broader reinforcement learning framework presents a promising opportunity to improve policy evaluation and learning in sequential decision-making problems. Finally, our paper primarily focuses on settings with small action spaces. Extending our approach to large action spaces is a promising direction, as importance weighting can break down in such cases due to extreme variance from large importance weights (Saito & Joachims, 2022).

## Acknowledgements

I sincerely thank the three anonymous reviewers for their insightful and constructive review comments and discussions, which substantially improved the manuscript. I also thank the Assigned Action Editor, Dr. Inigo Urteaga, and for the valuable discussions and guidance on the manuscript.

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

## Appendix

### The Proof of Proposition 3.2

Eqn. (10) directly follows

$$\mathbb{E}[\hat{f}^{\pi}(p_{ia})] - \pi_{ia}\mu_{ia} = \mathbb{E}[\Delta_{f}^{\pi}(p_{ia})] = O(n^{-q/(1+2q)}).$$

Thus, we have the bias

$$\mathbb{E}[\hat{V}_{\mathrm{nw}}^{\pi}] - V^{\pi} = n^{-1}\sum_{i=1}^{n}\sum_{a\in\mathcal{A}}\mathbb{E}\left[\Delta_{f}^{\pi}(p_{ia})\right] = O(Kn^{-q/(1+2q)}).$$

Noting $\mathbb{E}[(\hat{V}_{\mathrm{nw}}^{\pi} - \bar{V}_{\mathrm{nw}}^{\pi})(\bar{V}_{\mathrm{nw}}^{\pi} - V^{\pi})] = 0$ from (10), we have the mean squared error of $\hat{V}_{\mathrm{nw}}^{\pi}$:

$$\mathbb{E}[(\hat{V}_{\mathrm{nw}}^{\pi} - V^{\pi})^2] = \mathbb{E}[(\hat{V}_{\mathrm{nw}}^{\pi} - \bar{V}_{\mathrm{nw}}^{\pi})^2] + \mathbb{E}[(\bar{V}_{\mathrm{nw}}^{\pi} - V^{\pi})^2]. \tag{16}$$

For the first term of the righ-hand side of (16), we have

$$
\begin{aligned}
\mathbb{E}[(\hat{V}_{\mathrm{nw}}^{\pi} - \bar{V}_{\mathrm{nw}}^{\pi})^2] &\leq \left( n^{-1} \sum_{i=1}^{n} \mathbb{E}\left[ \sum_{a\in\mathcal{A}} \Delta_f^{\pi}(p_{ia}) \right]^2 \right) \\
&\leq \left( n^{-1} \sum_{i=1}^{n} |\mathcal{A}| \sum_{a\in\mathcal{A}} \mathbb{E}\left[ \Delta_f^{\pi}(p_{ia}) \right]^2 \right) \\
&= O(K^2 n^{-2q/(1+2q)}).
\end{aligned}
\tag{17}
$$

For the second term of the right-hand side of (16), $\bar{V}^{\pi}$ is a sample simple average of a set of sample with expectation $V^{\pi}$, following we have

$$
\mathbb{E}[(\bar{V}_{\mathrm{nw}}^{\pi} - V^{\pi})^2] = O(n^{-1}).
\tag{18}
$$

Inserting (17) and (18) to (16), the bound on the MSE in Proposition 3.2 is proved.

**The Proof of Proposition 4.1**

Similar to the proof of Proposition 3.2, we have

$$
\mathbb{E}[\hat{V}_{\mathrm{mnw}}^{\pi} - V^{\pi}] = O(Kn^{-q/(2q+1)}) + O(Kn^{-1}).
$$

For the MSE term, similar to the proof of Proposition 3.2, we have

$$
\begin{aligned}
\mathbb{E}[(\hat{V}_{\mathrm{mnw}}^{\pi} - V^{\pi})^2] &= \mathbb{E}[(\hat{V}_{\mathrm{mnw}}^{\pi} - \bar{V}_{\mathrm{mnw}}^{\pi})^2] + \mathbb{E}[(\bar{V}_{\mathrm{mnw}}^{\pi} - V^{\pi})^2]; \\
\mathbb{E}[(\bar{V}_{\mathrm{mnw}}^{\pi} - V^{\pi})^2] &= O(n^{-1})
\end{aligned}
$$

Now we investigate the term $\mathbb{E}[(\hat{V}_{\mathrm{mnw}}^{\pi} - \bar{V}_{\mathrm{mnw}}^{\pi})^2]$, and have

$$
\begin{aligned}
\mathbb{E}[(\hat{V}_{\mathrm{mnw}}^{\pi} - \bar{V}_{\mathrm{mnw}}^{\pi})^2] &= \mathbb{E}\left( n^{-1} \sum_{i=1}^{n} \sum_{a\in\mathcal{A}} \left[ \Delta_g^{\pi}(p_{ia}) + \pi_{ia}(\hat{\mu}_{ia} - \mu_{ia}^{*}) \right]^2 \right) \\
&\leq n^{-1}|\mathcal{A}| \sum_{i=1}^{n} \sum_{a\in\mathcal{A}} \mathbb{E}\left( \left[ \Delta_g^{\pi}(p_{ia}) + \pi_{ia}(\hat{\mu}_{ia} - \mu_{ia}^{*}) \right]^2 \right) \\
&\leq 2n^{-1}|\mathcal{A}| \sum_{i=1}^{n} \sum_{a\in\mathcal{A}} \left( \mathbb{E}\left[ \Delta_g^{\pi}(p_{ia}) \right]^2 + \mathbb{E}\left[ \pi_{ia}(\hat{\mu}_{ia} - \mu_{ia}^{*}) \right]^2 \right) \\
&= O(K^2 n^{-q/(1+2q)}) + O(K^2 n^{-1}).
\end{aligned}
$$

Thus, the MSE expression in the theorem follows.

**More results on the experiments**

**Performance under the estimated logging policy.**   We conduct an experiment where the logging policy $b$ is estimated. Unlike the previous experiment, the logging policy is estimated via logistic regression using four-fifths of the test data (Table 6), and policy value estimates are computed using various methods. We calculate the resulting bias and RMSE. We repeat this entire process 20 times, and report the average bias and RMSE values across these 20 runs in Table 5.

**Performance across evaluation sample sizes.** We examine performance across a range of evaluation sample sizes under the true logging policy. Figure 1 reports results for the Page data as a representative example. We report the RMSE along with its standard error, computed over 100 repetitions and displayed as error bars, as well as the bias and its standard error. The results show that our approach consistently outperforms competing methods across sample sizes.

Table 5: Bias and RMSE of various estimators for classification error under estimated logging policy.

| Data | Bias | | | | | RMSE | | | | |
|---|---|---|---|---|---|---|---|---|---|---|
| | DM | IPW | DR | NW | MNW | DM | IPW | DR | NW | MNW |
| letter | 0.503 | 0.006 | **0.000** | 0.001 | 0.009 | 0.503 | 0.030 | 0.035 | **0.029** | 0.034 |
| glass | 0.221 | 0.290 | -0.317 | -0.038 | **0.022** | 0.221 | 0.566 | 0.638 | **0.268** | 0.292 |
| ecoli | 0.197 | 0.115 | -0.208 | -0.009 | **0.003** | 0.197 | 0.305 | 0.417 | **0.112** | 0.128 |
| opt | 0.290 | 0.009 | -0.042 | **-0.001** | -0.004 | 0.290 | 0.036 | 0.062 | **0.023** | 0.030 |
| page | 0.090 | 0.009 | 0.001 | **0.000** | 0.010 | 0.090 | 0.020 | 0.033 | **0.011** | 0.017 |
| pen | 0.396 | 0.002 | -0.004 | **-0.001** | 0.003 | 0.396 | 0.018 | 0.025 | **0.017** | 0.023 |
| sat | 0.193 | 0.005 | 0.019 | **-0.001** | 0.022 | 0.193 | 0.021 | 0.027 | **0.019** | 0.029 |
| vehicle | 0.239 | 0.009 | **-0.002** | -0.005 | 0.012 | 0.239 | 0.054 | 0.051 | 0.045 | **0.044** |
| yeast | 0.215 | 0.015 | 0.065 | **0.000** | 0.069 | 0.215 | 0.084 | 0.097 | **0.074** | 0.095 |

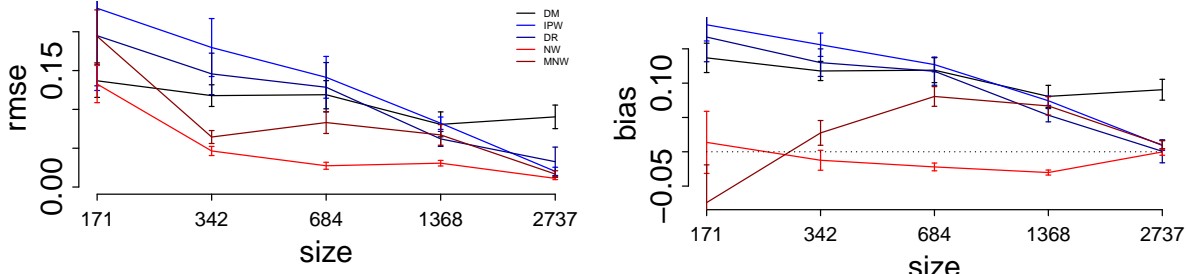

Figure 1: Performance across various sample size for the Page dataset. Left: RMSE as a function of sample size; Right: Bias as a function of sample size.

Table 6: Summary of the datasets, where $n$ denotes the total sample size, $d$ is the dimension of covariates, $n_v$ is the sample size of the evaluation, and $n_p$ is the sample size used to estimate the training behavior policy.

| Data | $n$ | $d$ | $n_v$ | $n_p$ |
|---|---|---|---|---|
| letter | 20000 | 16 | 10000 | 7500 |
| glass | 214 | 10 | 107 | 80 |
| ecoli | 336 | 8 | 168 | 126 |
| opt | 3823 | 64 | 1912 | 1434 |
| page | 5473 | 10 | 2737 | 2052 |
| pen | 7494 | 16 | 3747 | 2810 |
| sat | 6435 | 36 | 3218 | 2413 |
| vehicle | 846 | 18 | 423 | 317 |
| yeast | 1484 | 9 | 742 | 556 |

