# OpenReview forum: "From Weighting to Modeling: A Nonparametric Estimator for Off-Policy Evaluation"
_TMLR — Accepted by TMLR_

### Review · Reviewer_m39c · 2025-11-27

**Summary Of Contributions:**

This paper introduces a new approach for off-policy value estimation, called non-parametric weighting, using a P-spline approach. The paper compares this to two common weighting methods (inverse probability weighting and simple weighting) and shows the connection between them. The paper establishes some theoretical properties of NW, performing an error analysis to determine the rates of convergence for various quantities under usual assumptions. The paper also produces a model-assisted variant of NW and compares it to the DR estimator. Through experiments, the paper establishes the improved performance of the NW(/MNW) estimator compared to the other approaches.

**Additional Comments:**

In general, I am uncertain about the positioning of this work within the broader context of related work, and will defer to other reviewers to make judgements on this aspect (including whether the experiments are missing important baselines).

**Audience:**

Yes

**Audience Explanation:**

The problem of off-policy evaluation is a key challenge in the study of bandits and reinforcement learning. Improved off-policy evaluation for contextual bandits, which is directly addressed in this paper, could lead to improvements in various contextual bandit applications. Though not covered directly in this work, future works may build upon this work to apply the approach in reinforcement learning, when off-policy value estimates are required.

**Claims And Evidence:**

Yes

**Claims Explanation:**

- The introduction does not over-claim about the scope of the paper.
- The theoretical claims seem to be correct and well-supported.
    - I did not find the section "Robustness to the estimation of $p_{ia}$." to be very convincing.
- The empirical set-up seems reasonable, though has some weaknesses in my opinion:
    - The results presented are mostly snapshots with the dataset size chosen by the authors. It would be valuable to see comparisons across dataset sizes.
    - The experiment to determine "How much does estimating the logging policy affect the results?" uses a proxy estimate for $b$ rather than actually estimating $b$. It would be interesting to know what happens when $b$ is actually estimated in practice, because this may affect the relative performance of the approaches. Or, perhaps seeing how the metrics change when the noise scale $\mathbb{E}\left\[\delta\right\]$ changes.
    - "Including a larger set of methods would not necessarily provide additional insights, as our primary aim is to examine the relative performance and robustness of the NW-type estimators in comparison with the standard IPW framework." This is ok, but I still think it would be nice to make some comparisons to other existing methods, as this would help to determine whether NW estimators are actually worth using in practice (compared to existing techniques).
    - Some details are missing from experiments to be able to replicate them based on the paper.

**Requested Changes:**

**The following changes would aid in securing my recommendation, but are not necessarily critical.**

- I did not follow the argument in "Robustness to the estimation of $p_{ia}$." Perhaps it can be clarified/strengthened.
- The empirical results presented are mostly static snapshots computed with a fixed $n$. I would find the results more helpful if the work provided plots comparing the approaches at different values of $n$ (on the $x$-axis) for a few different values of $K$ (on different plots). Most of these plots could go into an appendix.
- It's not clear to me how many knots were used, or what degree spline was used in the empirical results.
- The results for "How much does estimating the logging policy affect the results?" should either be based on using an actually-estimated logging policy, or provide some more elaboration for why the Gaussian noise approach used is appropriate.



**The below changes are not critical for securing my recommendation, though I believe they would strengthen the work:**

- Most of the text is well-written and free of errors, but the first paragraph of the introduction requires some editing.
- There is a typo in the second line of pg 7
- Sec 3.5 should be titled "An illustrative example'
- I found it difficult to follow the procedure in Sec 3.5. I'm still not 100% sure I could replicate the procedure, though I think I understand the idea. In particular:
    - does $y^2_{iK} = y^2_{jK}$? I think the answer is no, and that you sample $\vec{y}_i$ from an $K$-dimensional normal distribution $\mathcal{N}(\vec{0}, I)$ for each $i$
    - What is meant by "We perform sampling without replacement"? I thought that we would sample one $y_{ia}$ from $\vec{y}_i$ at each $i$ so sampling with/without replacement would be irrelevant
- I would find it useful to have an illustrative example of MNW outperforming NW.
- A table comparing convergence rates of various quantities between the different methods would be useful.

---

> ### Author Response · Authors · 2026-01-01
>
> Thank you very much for very helpful comments. According to them, we have modified this paper. Please check the blue highlighted part in Section 3.6, Section 4.2, and Section 5.
>
> * on the illustrative example
>
> Yes, the rewards are generated for each context i. We have rewritten the reward-generation process for clarity. Following your suggestion, we also conducted a toy simulation in Section 4.2 to illustrate the potential advantage of the MNW estimator over the NW estimator. Please see these modifications.
>
>
> * on Experiments
>
> We conducted an experiment in which the logging policy is estimated; the corresponding performance results are reported in Table 5 in the appendix. The results show that our approaches perform consistently across different settings.
>
> We further examine performance under varying sample sizes and report results for a representative dataset, Page, in Figure 1 in the appendix. These results demonstrate that our approaches are robust to the sample size used for policy evaluation.
>
> We also added more details to replicate them. For example, we provide a summary of the datasets used for policy evaluation.
>
>
> * on Robust to the estimation of $p_{ia}$.
>
> In the revised paper, we provide additional discussion on robustness to behavior policy estimation in Section 3.4. We separately investigate the impact of estimation error and the impact of bias arising from model misspecification. In particular, for the latter, we explain why our approach is robust to behavior policy estimation.

---

### Review · Reviewer_mo5Z · 2025-12-05

**Summary Of Contributions:**

This paper considers the problem of offline policy evaluation for contextual bandit. A method is proposed that models the expected reward of a hypothetical policy pi using a univariate non-parametric function with input being the probability of action of a policy used to generate the data. Extension to doubly robust regimes is also considered

**Audience:**

Yes

**Audience Explanation:**

Offline policy evaluation is very important in sequential decision making problems such as RL which is at the heart of modern AI research. While this paper only studies contextual bandit, it is reasonable to believe that people who work on RL or related fields will find the method relevant.

**Claims And Evidence:**

Yes

**Claims Explanation:**

Mostly the math is solid. I only have one question regarding the motivation of the model, which I will detail later in my comment.

**Requested Changes:**

I only have one major question that I'm requesting the authors to elaborate on.

In Eq. (5), the expected reward of a policy pi on a context t, is modeled as a nonparametric function with input being the probability of seeing an action in the data generating process. I struggle to understand the motivation of this:

1. Why would pi_{ia} be on the left-hand side of this model? We know pi_{ia} (since this is the policy we want to evaluate). Can we simply fit a model of r_{ia} using p_{ia}, and then recover r_{ia}pi_{ia} by simply multiplying the estimated reward with pi? This way, we don't have to fit a different model for different policies pi.

2. More importantly, why would r_{ia} (or pi_{ia}r_{ia}) have anything to do with p_{ia}? p_{ia} is the probability of taking action a on context x_i in the policy that generates the data set. I don't see why this has anything to do with the actual reward of this particular action. For example, if I use a randomized policy to generate the data, then r_{ia} has abosolutely nothing to do with p_{ia}.

This shows that certain assumptions, either implicit or explicit, have to be placed to motivate this model. I would suggest making such assumptions explicit. This means that Eq. (5) needs to be discussed in much more details, preferably standalone sections, to convince readers that this is a suitable model.

2.1. Perhaps, there is some requirement that the exploration policy is good. In this way, if an action is taken with higher probability, then the expected reward of that action is higher, justfying some connection between r_{ia} and p_{ia}. It would be greatly appreciated if this can be made rigorous. Perhaps with a toy example showing / deriving such connections explicitly.

2.2. Perhaps, the alignment between the exploration policy and the evaluate policy is good, so that p_{ia} closely aligns with pi_{ia} and therefore pi_{ia}r_{ia} is closely related to p_{ia}. Again, formalizing this and some toy examples will be helpful.

Once the key model in Eq.(5) is well motivated, I think the idea of using univariate non-parametric estimates is reasonable. This is actually quite intereting that I believe will inspire ideas in other forms of offline evaluation problems.

---

> ### Author Response · Authors · 2025-12-23
> **Justification of the Modeling Approach**
>
> Thank you very much for your helpful comments. In response, we have revised the manuscript accordingly. Please see the blue-highlighted revisions in Sections 3.1.1 and 3.1.2. We would appreciate any further comments on these changes.
>
> * on the motivation
>
> We have presented our representation results for off-policy evaluation.
> Define
> \begin{equation}
> f^{\pi}(p_{ia})=\mathbb{E}[\pi_{ia}r_{ia} | p_{ia}].
> \end{equation}
> Based on this definition, we obtain the following representation results.
>
> Using the definition of $f^{\pi}(p_{ia})$ and the law of total expectation, we obtain
> \begin{equation}
> V^{\pi} = \mathbb{E}[p_{ia_i}^{-1}f^{\pi}(p_{ia_i})];
> \end{equation}
> and
> \begin{equation}
> V^{\pi}= \mathbb{E}\left[f^{\pi}(p_{ia})\right].
> \end{equation}
>
>
> The first result shows that $f^{\pi}(p_{ia})$ possesses a design-based representation analogous to the IPW estimator, and the second one shows that it also admit a model-based representation analogous to the DM estimator.
> If we instead define $f^{r}(p_{ia})=\mathbb{E}[r_{ia} | p_{ia}]$, the resulting quantity $\pi_{ia}f^{r}(p_{ia})$ does not inherit these representation properties.
>
> This is indeed the key point. Thank you very much for reminding that it should be emphasized explicitly.
>
>
> * on the model framework
>
> In the off-policy evaluation problem, the data are assumed to be collected under a mechanism in which the action assignment $a_i$ for unit $i$ relies only on $p_{ia}$. That is, conditional on $p_{ia}$, the action assignment $a_i$ is independent of $\pi_{ia}r_{ia}$.
> Therefore, from the representation result of $f^{\pi}(p_{ia})$, we have the model framework, which links $\pi_{ia}r_{ia}$ to $p_{ia}$ based on the data set $\mathcal{S}$.
> This provides an additional justification for why the dataset suffice to recover the representation of $f^{\pi}(p_{ia})$

---

> > ### Comment · Reviewer_mo5Z · 2026-01-29
> > **Follow-up question**
> >
> > I thank the authors' response to my questions. I am satisfied with answer to my first question 2.1. However, for question 2.2, I am still struggling to understand why p_{ia} has anything to do with pi_{ia}r_{ia}. For example, suppose I use a randomized policy (that takes actions uniformly at random regardless of states): p_{ia} would be the same for every i and a. How can a model possibly be built in this case, mapping p_{ia} to pi_{ia}r_{ia}? I feel there are still assumptions. I do believe that a toy example (possibly with a toy exploration policy) is needed here, to shed light on why p_{ia} has predictive power over r_{ia}pi_{ia}.

---

> ### Author Response · Authors · 2026-01-30
>
> Thank you very much!
>
> We assumed the question 2.2 is a sub-quesiton of question 2.1, so we didn't response to it. Sorry for this misunderstanding.
>
> Regarding why $p_{ia}$ can do something with $\pi_{ia}r_{ia}$, given the behavior policy $(p_{1a_1}, ..., p_{na_n})$, the mapping $f()$ produces a sequence $f(p_{1a_1}), ..., f(p_{na_n})$ that aims to approximate $(\pi_{1a_1}r_{1a_1}, ..., \pi_{na_n}r_{na_n})$. The functional form $f()$ reflects the relationship between $\pi_{ia}r_{ia}$ and $p_{ia}$ and is fitted using a nonparametric method.
>
> In the case of uniformly choice that you assumed, where all $p_{ia}$ are equal, the model degenerates to $f(p_{ia})=\mu$, with $\mu$ being a constant. In this setting, an efficient estimator is a simple average of $\pi_{ia_i}r_{ia_i}$. This corresponds to Case 2: Simple Weighting in Section 3.1.3.  We have clarified this correspondence for this kind of exploration policy in this case. Thank you very much for bringing this to our attention.
>
> In the latest revised version, we include a toy example illustrating that the relationship between $\pi_{ia}r_{ia}$ with the exploration policy $p_{ia}$  conforms to the function form of $f()$. Please see the end of Section 3.1.

---

### Review · Reviewer_dMcV · 2025-12-23

**Summary Of Contributions:**

This paper addresses off-policy evaluation in contextual bandits by proposing two novel estimation approaches that aim to reduce the high variance associated with inverse probability weighting (IPW) method.

The paper proposes two novel estimation approaches for off-policy evaluation in contextual bandits. The proposed approaches aim to reduce the high variance associated with the existing methods like inverse probability weighting (IPW). The authors propose nonparametric weighting (NW), where they model the relationship between policy-weighted rewards and behavior policy probabilities using a flexible nonparametric function (P-splines). In addition, building on nonparametric weighting, they propose model-assisted nonparametric weighting (MNW) method that incorporates reward predictions to further reduce variance, similar to the existing method known as doubly robust (DR) estimation.

The authors provide theoretical analysis of the NW and MNW estimators, establishing convergence rates for bias and mean squared error relative to the true policy value. The authors empirically evaluate NW and MNW on 9 multi-class classification datasets with bandit feedback and show robust results under two different conditions - with true logging policy (Table 2) and when it is estimated (Table 3). They compare the proposed methods against existing methods like direct method (DM), IPW and DR.  Overall, NW and MNW achieve lower variance in value estimation and maintain low bias while demonstrating robustness to probability estimation errors.

**Audience:**

Yes

**Audience Explanation:**

Yes, the findings of the paper would be of interest to researchers working on contextual bandits and off-policy evaluation. While the current work focuses on the bandit setting, the core methodology may have implications for the broader reinforcement learning community.

**Claims And Evidence:**

Yes

**Claims Explanation:**

In the abstract, the authors state - "To further reduce variance, we incorporate reward predictions—similar to the DR technique— resulting in the Model-assisted Nonparametric Weighting (MNW) approach. We show that MNW yields accurate value estimates when either the reward model or the behavior policy model is well specified." and in the related work section, the authors state "While the goal of our MNW approach is not to guarantee the standard doubly robust property, it explicitly models and mitigates the bias introduced by reward modeling". I find the claim made in the abstract to be a bit ambiguous, and I would like to request the authors to clarify the ambiguity.

Update - The authors have clarified the ambiguity and updated the paper.

**Requested Changes:**

1. Typo in page 5, 'demonstrates' should be 'demonstrate'.
2. Possible typo in page 7, the equation for the residual in the first paragraph might be wrong.
3. In paragraph 2 of introduction, 'DM and IPS' instead of 'DM and IPW'
4. In paragraph 1 of introduction, 'collect a set of data according a policy' -> 'according to a policy'
5. In paragraph 1 of introduction, 'were not receive' -> 'did not receive'
6. In section 3.1, 'there is no any relation' -> 'there is no relation'
7. While the authors discuss relevant related work in Section 6, the empirical comparisons are limited. I would like to see comparisons with more recent methods mentioned in the related work.

---

> ### Author Response · Authors · 2026-01-01
>
> Thank you very much for your suggestions. We have incorporated the changes in the revised manuscript.
>
> * on the statement in the abstract
>
> Thank you for pointing this out. We agree that the original statement was not appropriate. Our intention was to convey that, compared with the NW approach, the MNW estimator can explain additional outcome variation by incorporating a baseline model. However, it was an overstatement to claim that MNW possesses the standard doubly robust property.
>
> We have revised the abstract accordingly. In addition, following the suggestion of Reviewer m39c, we conducted a toy simulation to illustrate the potential advantage of MNW over the NW estimator. Please see these revisions.

---

> > ### Comment · Reviewer_dMcV · 2026-01-17
> >
> > Thank you for addressing the comments and incorporating the requested changes in your revised manuscript. I appreciate your clarification regarding the doubly robust property claim and the revision of the abstract to better reflect the properties of the MNW estimator. The addition of the toy simulation to illustrate MNW's advantages over NW is also a valuable contribution that should help readers better.

---

### Decision · Action_Editor_jNU1 · 2026-02-16

**Recommendation:** Accept as is

**Audience:**

Yes

**Audience Explanation:**

Off-policy evaluation is a fundamental area of study within machine learning, and reinforcement learning in particular.

Although this work is limited to the contextual bandit setting, the demonstrated improvements in estimation ---specifically the reduction in variance--- will be of clear interest to the TMLR audience.

Besides, the potential to integrate these proposed techniques into broader RL frameworks offers a promising direction that the community may find valuable.

**Claims And Evidence:**

Yes

**Claims Explanation:**

The authors' work is based on a representation result for the value function in contextual-bandit off-policy evaluation.

As noted by reviewers, while the methodological contribution is incremental, the paper is solid and provides sufficient empirical evidence for its claims. This was especially true after the clarifications and changes introduced during the discussion period.

Specifically, the experiments effectively support the claim that the proposed nonparametric value estimator maintains low bias while achieving lower variance when compared to classic inverse probability weighting (IPW) and doubly robust (DR) estimators.